# Cognitive and Adaptive Characterization of Children and Adolescents with KBG Syndrome: An Explorative Study

**DOI:** 10.3390/jcm10071523

**Published:** 2021-04-06

**Authors:** Paolo Alfieri, Cristina Caciolo, Giulia Lazzaro, Deny Menghini, Francesca Cumbo, Maria Lisa Dentici, Maria Cristina Digilio, Maria Gnazzo, Francesco Demaria, Virginia Pironi, Giuseppe Zampino, Antonio Novelli, Marco Tartaglia, Stefano Vicari

**Affiliations:** 1Department of Neuroscience, Child and Adolescent Psychiatry Unit, Bambino Gesù Children′s Hospital, IRCCS, 00146 Rome, Italy; cristina.caciolo@opbg.net (C.C.); giulia.lazzaro@opbg.net (G.L.); deny.menghini@opbg.net (D.M.); francesca.cumbo@opbg.net (F.C.); francesco.demaria@opbg.net (F.D.); stefano.vicari@opbg.net (S.V.); 2Department of Human Science, LUMSA University of Rome, 00193 Rome, Italy; 3Genetics and Rare Diseases Research Division, Bambino Gesù Children’s Hospital, IRCCS, 00146 Rome, Italy; marialisa.dentici@opbg.net (M.L.D.); mcristina.digilio@opbg.net (M.C.D.); maria.gnazzo@opbg.net (M.G.); marco.tartaglia@opbg.net (M.T.); 4Center for Rare Disease and Congenital Defects, Fondazione Policlinico Universitario A. Gemelli, Catholic University, 00168 Rome, Italy; virginia.pironi@gmail.com (V.P.); giuseppe.zampino@unicatt.it (G.Z.); 5Laboratory of Medical Genetics, Bambino Gesù Children’s Hospital, IRCCS, 00146 Rome, Italy; antonio.novelli@opbg.net; 6Department of Life Sciences and Public Healt, Fondazione Policlinico Universitario A. Gemelli, Catholic University, 00168 Rome, Italy

**Keywords:** *ANKRD11*, 16q24.3, cognitive abilities, adaptive functioning, intellectual disabilities

## Abstract

KBG syndrome (KBGS) is a rare Mendelian condition caused by heterozygous mutations in *ANKRD11* or microdeletions in chromosome 16q24.3 encompassing the gene. KBGS is clinically variable, which makes its diagnosis difficult in a significant proportion of cases. The present study aims at delineating the cognitive profile and adaptive functioning of children and adolescents with KBGS. Twenty-four Italian KBGS with a confirmed diagnosis by molecular testing of the causative *ANKRD11* gene were recruited to define both cognitive profile as measured by the Wechsler Intelligence Scale and adaptive functioning as measured by Vineland Adaptive Behavior Scales-II Edition or the Adaptive Behavior Assessment System-II Edition. Among children and adolescents, 17 showed intellectual disability, six presented borderline intellectual functioning and only one child did not show cognitive defects. Concerning cognitive profile, results revealed significant differences between the four indexes of Wechsler Intelligence Scale. Namely, the verbal comprehension index was significantly higher than the perceptual reasoning index, working memory index and the processing speed index. Concerning adaptive functioning, no difference between the domains was found. In conclusion, in our cohort, a heterogeneous profile has been documented in cognitive profiles, with a spike on verbal comprehension, while a flat-trend has emerged in adaptive functioning. Our cognitive and adaptive characterization drives professionals to set the best clinical supports, capturing the complexity and heterogeneity of this rare condition.

## 1. Introduction

KBG syndrome (KBGS) (MIM #148050) is a rare Mendelian condition caused by heterozygous mutations in *ANKRD11* [1] or microdeletions encompassing the gene [2,3,4]. More than 100 affected individuals have been reported in the literature; however, it is likely that the syndrome is underdiagnosed due to mild features. Features that are typically present at birth may be difficult to recognise until developmental delays are apparent, or permanent teeth erupt. Moreover, it is likely that this syndrome is less frequently diagnosed since features are not severe and fairly common among other disorders [5]. Main features of KBGS include short stature, distinctive facial appearance, macrodontia of the permanent central upper incisors, cardiac defects, palate abnormalities, sleep disturbances, feeding difficulties, hearing problems, speech delay, and learning difficulties [1,6,7,8,9,10,11,12]. Affected patients may not show the characteristic KBG phenotype. Indeed, reverse phenotyping documented that the diagnosis of KBGS can be missed because of the wide heterogeneity of phenotypic manifestations [11,12]. Based on these considerations, Low and colleagues [11] updated diagnostic criteria/clinical recommendations for this rare disorder, proposing that a diagnosis of KBGS should be considered in patients with developmental delay/learning difficulties, speech delay or significant behavioural issues with at least two major criteria (e.g., macrodontia, height below 10th centile) or one major plus two minor criteria (e.g., brachydactyly, seizures). Notably, conclusive phenotype–genotype correlations have not been identified [11,12].

However, accurate cognitive profiling of subjects with mutated *ANKRD11* alleles is still lacking. Currently, anecdotal evidence suggests that individuals with KBGS usually present a high variability of intellectual impairment, often ranging from mild to moderate [13,14,15], even though a borderline to normal range cognition has also been reported [15,16,17]. Most studies refer to case reports, and only two surveys have robustly described the cognitive profile of the disorder [14,18]. The latter reported that around 10% of children and adults with KBGS show moderate intellectual disability (ID), with approximately 50% of cases having mild ID, while the remaining are characteried by borderline to normal IQ, as assessed by the Wechsler Intelligence Scale for Children Third Edition [14,18]. 

With the introduction of the *Diagnostic and Statistical Manual of Mental Disorders* (DSM-5) [19], the evaluation of adaptive behaviour has become a necessary requirement in order to establish the severity of the ID. It is now largely accepted that clinicians should consider adaptive behaviour, which “*represents the conceptual, social, and practical skills that people have learned to be able to function in their everyday lives*” [20]. Current knowledge on cognitive behaviour associated to adaptive assessment in KBGS is still sparse and has been barely investigated. To the best of our knowledge, only a case-report study included the adaptive profiling in the evaluation of two individuals with KBGS, showing a low average intellectual level with severe impairment in the communication domain of adaptive behaviour [13]. 

The current explorative study aims at defining both cognitive profile and adaptive behaviour, respectively assessed by Wechsler Intelligence Scales [21,22] and adaptive behaviour measures (Vineland Adaptive Behavior Scales-Second Edition-Survey Interview Form-VABS-II-SIF [23] or Adaptive Behavior Assessment System-Second Edition-ABAS-II [24]) in patients with a clinical diagnosis of KBG syndrome, confirmed by the identification of a *ANKRD11* pathogenic mutation or with a 16q24.3 microdeletion. Specifically, our overarching goal was to more accurately characterise the cognitive profile in a cohort of Italian children and adolescents with KBGS, considering both intellectual levels and adaptive behaviour by using “gold standard” measures currently administered for diagnosis of ID [19].

Based on our clinical experience, we predicted a heterogeneous cognitive profile in KBGS, with strong verbal comprehension abilities compared to perceptual reasoning, working-memory and speed processing, but with a flat-trend in adaptive behaviour. 

## 2. Materials and Methods

### 2.1. Participants

Our cohort was composed by 24 Italian children and adolescents with molecularly confirmed diagnosis of KBGS (M/F = 14/10; mean age: 11.99 ± 3.93 years). Participants were recruited at the Child and Adolescent Psychiatry Units of Bambino Gesù Hospital and Policlinico Universitario Agostino Gemelli in Rome. Fifteen participants had already enrolled in a previous related study [25].

All participants and their parents gave their informed consent in accordance with the Code of Ethics of the World Medical Association (Declaration of Helsinki), and the study was approved by the Bambino Gesù Children′s Hospital Ethical Committee (protocol code 1094_OPBG 1940/2019 03-07-2019, Protocol number 1094, on 3 July 2019).

#### 2.1.1. Molecular Analysis

The patient′s genomic DNA was extracted from circulating leukocytes using QIAampH DNA Blood Kit (QIAGEN Sciences, Germantown, MD, USA) according to the manufacturer′s instructions.

Comprehensive open reading frame/splice site mutational analysis of all ANKRD11 exons (16p24.3; GRCh38: chr16:89,267,618-89,490,560, 9301 bp) was performed through targeted resequencing, using a customised panel, Nextera technology (Illumina) or SeqCap EZ Choice Enrichment Kit (Roche), and analysed on a MiSeq sequencing platform or NextSeq550 (Illumina, San Diego, CA, USA).

All genetic variants detected in index cases were validated on re-extracted DNA by bidirectional Sanger sequencing, using the standard protocol with the BigDye Terminator v3.1 Cycle Sequencing Kit, separated on the ABI Prism 3130xl Genetic Analyzer (Applied Biosystem), and analysed by Mutation Surveyor DNA Variant Analysis Software (Softgenetics).

Potentially pathogenic rare variants (MAF  <  1%) were explored in the Ensembl genome browser (http://www.ensembl.org/, accessed on 1 February 2021), the 1000 Genomes Project, National Center for Biotechnology Information (NCBI, http://www.ncbi.nlm.nih.gov/, accessed on 1 February 2021), Exac Browser (http://exac.broadinstitute.org/, accessed on 1 February 2021) and gnomAD Browser (https://gnomad.broadinstitute.org/, accessed on 1 February 2021). Array-CGH (Comparative Genomic Hybridization) analysis at a resolution of 100 kb was performed in patients in which the molecular analysis of *ANKRD11* was negative. Further information is available upon request.

#### 2.1.2. Clinical Examination

Children and adolescents with KBGS were assessed by experienced developmental psychiatrists for mental disorders according to developmental history and extensive clinical examination. According to DSM-5 criteria, of the 24 children and adolescents, 6 presented borderline intellectual functioning, 6 presented mild ID, 5 presented mild/moderate ID, and the rest presented moderate ID. Only one child did not present ID, but received a diagnosis of developmental coordination disorder.

All individuals underwent a detailed evaluation directed to define cognitive profile and adaptive functioning.

Cognitive profile was assessed using the Italian version Wechsler Intelligence Scale for Children-Fourth Edition (WISC-IV) [21] and, for one individual, the Italian version of the Wechsler Adult Intelligence Scale-Fourth Edition (WAIS-IV) [22]. Both WISC-IV and WAIS-IV generate Full IQ, VCI, PRI, WMI, and PSI scores. VCI is an overall measure of the child′s ability to verbally reason, PRI reflects an individual′s ability to accurately interpret, organise and think with visual information, while WMI is a measure of working memory ability and PSI is a measure of processing speed. The Full-IQ and each index yield age-based standard scores (mean, M = 100, standard deviation, SD = 15).

Adaptive functioning was assessed using two of the most widely used scales, considered the gold standards to investigate the conceptual, social and practical domain to which DSM-5 refers: the VABS-II-SIF [23] and Italian version of Adaptive Behavior Assessment System-Second Edition Parent Form 5-21 [24].

VABS-II-SIF assesses adaptive functioning of individuals from birth to 90 years and 11 months through caregivers′ interviews, and yields three domain scores: Communication, Socialization and Daily Living Skills (the fourth Motor Skills domain is investigated only for children younger than 7 years). An overall Adaptive Behavior Composite score is also provided. The VABS-II-SIF Adaptive Behavior Composite and relative domains yield age-based standard scores (M = 100, SD = 15). 

The ABAS-II 5-21 is a questionnaire that the caregivers complete for children, adolescents and young adults from 5 to 21 years. ABAS-II 5-21 yields three specific domain scores (Conceptual, Social, and Practical) and an overall General Adaptive Composite. Domains have age-based standard scores (M = 100, SD = 15).

Although we used different tools for the adaptive functioning, it was possible to combine the scores referring to similar domains. Indeed, in a study reported in the manual of the Vineland II Italian version [23], the correlations between VABS-II-SIF and ABAS-II 5-21 were run. The correlations between similar domains were high (between 0.60 and 0.74): Communication correlated significantly and positively with the Conceptual domains (r = 0.68), Socialization correlated significantly and positively with the Social domains (r = 0.60) and Daily Living Skills correlated significantly and positively with the Practical domains (r = 0.74). In addition, the correlation between Adaptive Behavior Composite score of VABS-II-SIF and General Adaptive Composite (hereinafter in the text: Full-Adaptive Scale), was 0.78.

Table 1 depicts demographic features (age, gender), cognitive abilities (Full-IQ, Verbal Comprehension Index (VCI), Perceptual Reasoning Index (PRI), Working Memory Index (WMI), Processing Speed Index (PSI)), and adaptive behaviours (Full-Adaptive Scale, Communication, Socialization, Daily Living Skills).

### 2.2. Statistical Analysis

Descriptive statistics (mean, min–max, SD and confidence interval at 95%) were calculated for scores of cognitive abilities as well as those related to adaptive behaviour. 

To detect strengths and weaknesses within subcomponents of cognitive abilities in children and adolescents with KBGS, a multivariate analysis of variance (MANOVA) was conducted with Wechsler scale indexes (VCI, PRI, WMI, PSI) as within-subject factors. Further, to explore strengths and weaknesses within adaptive functioning in KBGS, the same statistical analysis was used to test differences between the domains of adaptive behaviour scale (Communication, Daily living skills, Socialization). 

Delta (Δ) between scores of each index (VCI, PRI, WMI, PSI) and Full-IQ were calculated (respectively, Δ_VCI_ = VCI − Full-IQ, Δ_PRI_ = PRI − Full-IQ, Δ_WMI_ = WMI − Full-IQ, Δ_PSI_ = PSI − Full-IQ). To verify the supremacy of verbal comprehension abilities compared to the general cognitive abilities, a MANOVA was conducted with Δ scores of each index (Δ_VCI_, Δ_PRI_, Δ_WMI_, Δ_PSI_) as within-subject factors. 

Post hoc analyses were performed using Fisher′s LSD test. Partial eta square (η*_p2_*) has been reported as an effect size measure. A *p* value of less than 0.05 was considered as statistically significant.

To investigate the supremacy of verbal comprehension abilities compared to the other Wechsler indexes from a more clinical perspective, the statistical differences between VCI and PRI, WMI, PSI (VCI vs. PRI; VCI vs. WMI; VCI vs. PSI) in our cohort and those of the general population were counted per individual according to the significance tables from the normative data manual [21,22]. The percentage (%) of individuals who showed at least one significant discrepancy between indexes (VCI vs. PRI; VCI vs. WMI; VCI vs. PSI) was calculated.

## 3. Results

### 3.1. Molecular Results

Genetic variants identified in our patients are shown in Table 2. All mutations affected the C-terminal region at exon 9 of ANKRD11, except one mutation affecting exon 10 of the gene and two small 16q24.3 microdeletions involving *ANKRD11* without affecting adjacent genes that were detected by array-CGH analysis in single individuals. The mutations included 16 frameshift, 5 nonsense and 1 missense mutation. Microdeletions and 19 of the 22 intragenic mutations were novel. In one case, the pathogenic variant had been inherited from the affected mother. In four families, unaffected parents were not available for analysis (Table 2). Further details on relevant medical features of our cohort are reported in Appendix A (see Appendix A).

### 3.2. Cognitive Abilities and Adaptive Behavior

Descriptive statistics for cognitive abilities of our cohort are depicted in Table 3. The results of the MANOVA revealed significant differences between the indexes of Wechsler scales (F_3,69_ = 4.83, *p* = 0.004, η*_p_*^2^ = 0.17). As shown in Figure 1 (panel a), the mean score of VCI was significantly higher than those of the PRI (*p* = 0.013), WMI (*p* = 0.001) and PSI (*p* = 0.003). No other significant differences were documented (PRI vs. WMI: *p* = 0.40; PRI vs. PSI: *p* = 0.58; WMI vs. PSI: *p* = 0.77).

Concerning Δ scores of each index, the results of the MANOVA were found to be significant (F_3,69_ =4.83, *p* = 0.004, η*_p_*^2^ = 0.17). Namely, the mean Δ score of VCI was seen to be significantly higher than the Δ scores of the other indexes (Δ_VCI_ vs. Δ_PRI_: *p* = 0.013, Δ_VCI_ vs. Δ_WMI_: *p* = 0.001, Δ_VCI_ vs. Δ_PSI_: *p* = 0.003). No additional differences between Δ scores were observed (all comparisons *p* > 0.10) (see Figure 2).

Descriptive statistics for *adaptive behaviour* in our children and adolescents with KBGS are depicted in Table 4. No differences between the domains of adaptive behaviour scale (F_2,46_ = 1.45, *p* = 0.25, η*_p_*^2^ = 0.06) were found (Figure 1b). 

#### Statistical Differences between VCI and Other Index 

In accordance with the significance tables from the normative data manual, we found that almost 30% of our sample presented statistical differences between VCI and at least one of the remaining indexes (PRI, WMI, PSI). Please refer to Table 5.

## 4. Discussion

Our overarching goal was to characterise patterns of cognitive profile and adaptive behaviour in a clinically characterised cohort of 24 children and adolescents with KBGS in which a clinical diagnosis was confirmed by molecular testing, with the detection of 22 pathogenic variants in *ANKRD11* gene and two 16q24 microdeletions encompassing the *ANKRD11* gene, entirely. To the best of our knowledge, the current study first revealed a heterogeneous profile within subcomponents of cognitive abilities tested by Wechsler scales′ indexes and a flat-trend in the domains of adaptive functioning tested by “gold standard” measures for clinical routines. 

Concerning cognitive profile, our cohort documented a wide variability of intellectual levels (from severely below average to normal range) with a mean score of Full-IQ falling around 2 SDs below average. Specifically, 25% of children and adolescents with KBGS presented a Full-IQ below 3 SDs compared to the control population, and 33% between −3 and −2 SDs below the mean. Only 9% of the tested cohort showed Full-IQ scores above −1 SD. These findings confirm previously collected data indicating a high variability of intellectual levels in individuals with KBGS [5,13,15,16,17].

Even if Full-IQ is usually informative of global intellectual functioning, from a clinical point of view, it is also crucial to investigate the four index scores (VCI, PRI, WMI, and PSI) related to the broad cognitive abilities, each of which contributes equally to determine the global intellectual functioning. Intriguingly, as predicted, a more in depth exploration of these indexes showed higher scores in VCI compared to PRI, WMI, PSI. When comparing the differences of each broad cognitive index from Full-IQ (Δ_VCI_, Δ_PRI_, Δ_WMI_, Δ_PSI_), a spike emerged only in the score of VCI, since the mean score deviated around 1 SD above the average score of Full-IQ. Surprisingly, when comparing to normative data of general population, 30% of our cohort presented a statistical difference between VCI and at least one of the remaining indexes. In sum, although our cohort of children and adolescents with KBGS presented a below-average intellectual level, strong verbal comprehension abilities were documented. 

When considering cognitive abilities, our data are partially discordant to previous studies. Indeed, the mean Full-IQ score of our cohort is in line with the previous findings [14,18]. While different from what was reported by van Dongen and colleagues [14], we did not observed a flat-trend among verbal comprehension, perceptual reasoning, working memory and speed of information processing abilities. Comparing our results to the latter study is difficult, due to dissimilarities between the design of the tools used as well as to the intelligence models at the basis of them (WISC-III, two-factor intelligence model [28,29] vs. WISC-IV/WAIS-IV, Cattell–Horn Carroll Theory of Cognitive Abilities [30]). 

Concerning adaptive functioning, mean scores across domains were observed to fall over 2 SDs below the average, showing overall impaired adaptive behaviours. Specifically, 35% of children and adolescents with KBGS presented a Full-Adaptive Scale below 3 SDs from the mean, 39% between −3 SDs to −2 SDs below the mean, and only 4% of the tested subjects had scores above −1 SD. In our sample, the majority of patients (74%) showed a significant impairment (at least −2 SDs below the mean) in adaptive functioning. While considering only Full-IQ, the same impairment was present only in 58% of patients. This again underlines the importance of ensuring a detailed clinical examination to better define degrees of global severity for each patient. Exploring domains of adaptive behaviour, the results confirm our exploratory hypotheses showing a flat-trend. Although significant differences did not emerge, the communication domain resulted slightly lower than daily living skills and socialization domains. This finding was aligned to a previous study [12], which described adaptive skills in two patients. In this small study, the authors revealed severe impairments in written and narrative communication abilities despite the relative strength in domestic and socialization skills. Surprisingly, our data indicate a high VCI score in Wechsler Scales that apparently does not match with the communication domain in the adaptive profile, which appeared to be more compromised. This result is difficult to interpret and deserves to be explored in a larger sample and by a deeper neuropsychological assessment. 

The distribution of diagnosis of ID in our cohort was partially discordant with those reported in a previous work [14] documenting a lower proportion of patients with moderate ID. This imbalance of ID distribution between our study and that of van Dongen [14] could be related to the influence of adaptive behaviour in determining the degree of ID. In fact, taking together intellectual levels and adaptive behaviour, according to the diagnostic criteria of DSM-5 [19], the clinical diagnosis of ID followed this distribution in our patients: 29% presented from normal to borderline intellectual functioning, 25% presented Mild ID, and 46% presented from Mild/Moderate ID to Moderate ID. 

Even if focusing on Full-IQ and Full-Adaptive Scale drives clinicians in the diagnostic process, observing strengths and weaknesses of both cognitive profile and adaptive functioning is equally important in this rare syndrome. This allows researchers to better characterise the cognitive phenotype and helps clinicians to set up highly specific rehabilitative treatments and ad hoc supports. The importance of an in-depth cognitive and adaptive characterization was already emphasised in other rare syndromic conditions (i.e., Williams syndrome, Down syndrome). For example, considering cognitive abilities of individuals with Williams syndrome, they typically show a clear strength in language, and an extreme weakness in visuospatial construction. The adaptive behaviour profile seems to be characterised by clear strength in socialization skills, strength in communication, and clear weakness in daily living and motor skills [31]. Instead, considering Down syndrome, studies focusing on the cognitive characteristics reveal relative weaknesses in expressive language, syntactic/grammar processing, verbal working memory [32], and fewer adaptive behavioural problems than those of individuals with other intellectual disabilities [33].

By looking at the results of our cohort, we can speculate that if neuropsychiatric team assesses a patient with high verbal comprehension abilities and flat adaptive behaviour in all domains in addition to some physical characteristics [11], they could assume to perform a genetic deepening for KBGS. Of importance, the need to better define components of the cognitive abilities and adaptive behavior for the study of genotype/phenotype correlations in KBGS is stressed here.

One limitation of this study is the lack of a control group: our analyses focused on comparisons with normative data. Future studies should be designed to include a control group and enlarge the sample size to obtain more consistent data and to better characterise the cognitive profile in KBGS. Further studies should follow the evolution and relation of cognitive and adaptive abilities longitudinally, in order to delineate the developmental trend of this rare genetic condition.

Summing up, our findings stressed the importance of meticulously evaluating children and adolescents with KBGS following two simultaneous clinical approaches. On one hand, professionals could follow an analytic approach, aiming at carefully investigating cognitive and adaptive abilities considering general but also subcomponents as indexes or domains. Indeed, at first glance, if only Full-IQ was considered, our sample would have presented a mild cognitive impairment. Deeper focusing on each index allowed us to delineate both strengths and weakness, otherwise hidden from a general cognitive ability. 

On the other hand, clinicians should focus not only on intellectual level but also on adaptive functioning, to determine the degree of ID according to DSM-5 criteria. Indeed, we recommend the assessment of adaptive functioning, also when Full-IQ is not below the average.

Overall, if future studies with larger sample sizes will confirm heterogeneous profile, we think that extensive observation of different cognitive patterns (i.e., verbal comprehension abilities, perceptual reasoning, working memory abilities and speed of information processing), could potentially represent a diagnostic aid for clinical use in KBGS, as previously reported for other features [11]. 

In conclusion, our in-depth cognitive and adaptive characterization drives professionals to set the best clinical supports, capturing the complexity and heterogeneity of this rare condition.

## Figures and Tables

**Figure 1 jcm-10-01523-f001:**
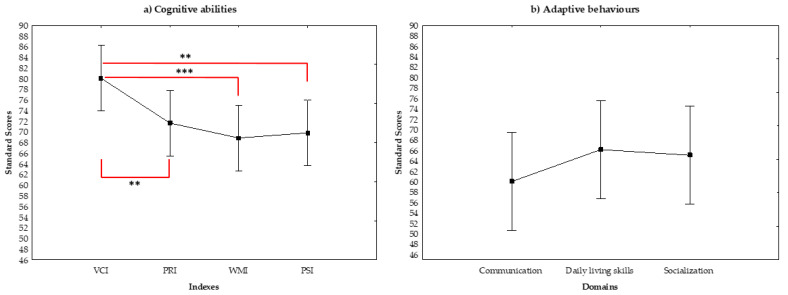
Panel **a**, **b** Mean of standard scores in Wechsler scale (**a**) and domains of adaptive behaviour (**b**). VCI verbal comprehension index; PRI perceptual reasoning index; WMI working memory index; PRI processing speed index; ** *p* ≤ 0.01; *** *p* ≤ 0.001.

**Figure 2 jcm-10-01523-f002:**
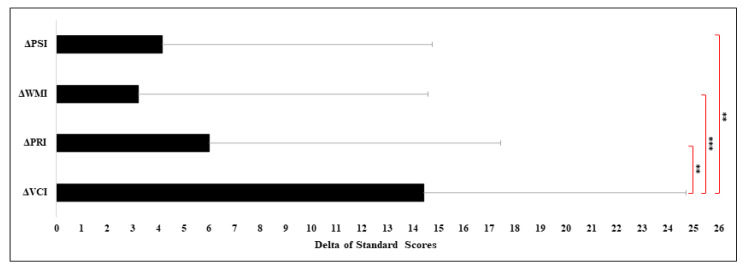
Δ Indexes of Wechsler Scale in our cohort of KBGS (KBG syndrome). Legend: Δ_VCI_: delta verbal comprehension index (VCI − Full-IQ); Δ_PRI:_ delta perceptual reasoning index (PRI − Full-IQ); Δ_WMI_: delta working memory index (WMI − Full-IQ); Δ_PSI:_ processing speed index (PSI − Full-IQ). ** *p* ≤ 0.01; *** *p* ≤ 0.001.

**Table 1 jcm-10-01523-t001:** Demographic features, cognitive abilities, and adaptive behaviour of studied cohort.

*N*	Age	Gender	Wechsler Scale					Adaptive Scale			
			Full-IQ	VCI	PRI	WMI	PSI	Full-Adaptive Scale	COM	DLS	SOC
1	15.83	M	69	78	87	73	65	23	20	69	20
2	9.56	F	63	94	63	61	65	46	51	49	57
3	10.40	M	80	88	76	85	91	95	95	94	97
4	15.30	M	49	72	50	58	68	56	61	42	78
5	15.12	F	47	70	61	58	47	20	20	22	21
6	23.91	M	42	51	63	52	42	20	20	47	20
7	9.51	M	89	98	91	67	103	50	61	45	66
8	11.65	F	82	84	100	76	82	70	72	73	74
9	11.54	F	81	96	65	73	85	85	86	89	84
10	12.27	M	68	70	76	79	82	67	72	75	66
11	8.99	F	57	86	52	70	59	73	83	61	82
12	14.0	F	48	64	52	64	65	27	45	55	20
13	7.24	M	71	76	71	46	82	52	49	65	60
14	13.58	M	76	92	74	79	82	81	83	85	87
15	13.75	M	43	58	71	46	47	35	36	47	56
16	10.83	F	84	124	74	88	56	74	93	58	72
17	8.0	M	77	88	102	67	67	67	77	98	81
18	10.42	F	66	76	58	88	79	67	54	41	60
19	6.1	F	90	112	87	85	79	70	74	84	90
20	17.33	M	58	69	65	63	75	72	36	98	97
21	9.0	M	58	68	74	62	58	66	63	84	67
22	12.4	M	57	78	65	64	59	67	64	83	68
23	14.0	F	42	58	56	58	53	59	66	58	74
24	7.0	M	79	72	87	91	85	n.a.	n.a.	n.a.	n.a.

Legend: M, male; F, female; Full-IQ, full intelligence quotient; VCI, Verbal Comprehension Index; PRI, Perceptual Reasoning Index; WMI, Working Memory Index; PSI, Processing Speed Index; COM, Communication; DLS, Daily Living Skills; SOC, Socialization; n.a., not available.

**Table 2 jcm-10-01523-t002:** List of ANKRD11 variants identified in our cohort/Genotypic summary of our KBG patients.

Subject	Genomic Coordinate	Nucleotide Position	Protein Position	Exon	dbSNP	gnomAD	Mutation Type	ClinVar ID	Segregation	ACMG Classification	Reference
1	chr16:89350973	c.1977C > G	p.Tyr659 *	9	rs749201074	-	nonsense	489328	de novo	Pathogenic	[12]
2	chr16:89350772	c.2175_2178delCAAA	p.Asn725Lysfs * 23	9	rs886039734	0.000003993	frameshift	265689	de novo	Pathogenic	[12]
3	chr16:89347745	c.5205delC	p.Val1736Cysfs * 227	9	-	-	frameshift	-	de novo	Pathogenic	[12]
4	chr16:89345758	c.7192C > T	p.Gln2398 *	9	rs1265287370	-	nonsense	-	de novo	Pathogenic	[12]
5	chr16:89350538	c.2412delA	p.Glu805Lysfs * 58	9	rs886039902	-	frameshift	-	maternal (affected mother)	Pathogenic	[12]
6	chr16:89346866	c.6071_6084del14	p.Pro2024Argfs * 3	9	-	-	frameshift	-	de novo	Pathogenic	This study
7	chr16:89345534	c.7416C > G	p.Tyr2472 *	9	-	-	nonsense	-	de novo	Pathogenic	[12]
8	chr16:89349179	c.3770_3771delAA	p.Lys1257Argfs * 25	9	rs886039477	-	frameshift	265324	de novo	Pathogenic	[12]
9	chr16:89348560	c.4389_4390delGA	p.Lys1464Thrfs * 89	9	rs1597451815	-	frameshift	817640	de novo	Pathogenic	[26]
10	chr16:89283689	chr16:89283689_89572450 deletion		entire gene	-	-	microdeletion	-	de novo	Pathogenic	[12]
11	chr16:89351043	c.1903_1907delAAACA	p.Lys635Glnfs * 26	9	rs886041125	-	frameshift	279678	de novo	Pathogenic	[26]
12	chr16:89351664	c.1285_1286delTC	p.Ser429Glyfs * 8	9	rs1597465419	-	frameshift	633543	de novo	Pathogenic	[12]
13	chr16:89351043	c.1903_1907delAAACA	p.Lys635Glnfs * 26	9	rs886041125	-	frameshift	279678	de novo	Pathogenic	[26]
14	chr16:89350772	c.2175_2178delCAAA	p.Asn725Lysfs * 23	9	rs886039734	0.000003993	frameshift	265689	de novo	Pathogenic	[12]
15	chr16:89350549	c.2398_2401delGAAA	p.Glu800Asnfs * 62	9	rs797045027	-	frameshift	209131	de novo	Pathogenic	[27]
16	chr16:89347238	c.5712_5713insT	p.Gly1905Trpfs * 45	9	-	-	frameshift	-	parents not tested	Pathogenic	This study
17	chr16:89283689	chr16:89283689_89559189 deletion		entire gene	-	-	microdeletion	-	de novo	Pathogenic	This study
18	chr16:89347806	c.5144dupA	p.Tyr1715 *	9	-	-	nonsense	-	de novo	Pathogenic	This study
19	chr16:89349641	c.3309dupA	p.Asp1104Argfs * 2	9	rs772267579	0.000007970	frameshift	812782	de novo	Pathogenic	[12]
20	chr16:89348452	c.4498C > T	p.Gln1500 *	9	-	-	nonsense	-	parents not tested	Pathogenic	[12]
21	chr16:89351566	c.1381_1384delGAAA	p.Glu461Glnfs * 48	9	rs1597464953	-	frameshift	633578	parents not tested	Pathogenic	[17]
22	chr16:89349356	c.3591_3594delAAAA	p.Lys1198Argfs * 119	9	-	-	frameshift	-	de novo	Pathogenic	This study
23	chr16:89351566	c.1381_1384delGAAA	p.Glu461Glnfs * 49	9	rs1597464953	-	frameshift	633578	parents not tested	Pathogenic	[17]
24	chr16:89341503	c.7567C > T	p.Arg2523Trp	10	-	-	missense	-	de novo	Likely pathogenic	This study

**Table 3 jcm-10-01523-t003:** Cognitive abilities of children and adolescents with KBG syndrome.

Wechsler Indexes	Mean	Min–Max	SD	CI 95%
Full-IQ	65.67	42–90	15.44	12.00–21.66
Verbal Comprehension Index	80.08	51–124	17.14	13.32–24.04
Perceptual Reasoning Index	71.67	50–102	14.64	11.38–20.54
Working Memory Index	68.88	46–91	13.05	10.14–18.31
Processing Speed Index	69.83	42–103	15.52	12.07–21.78

**Table 4 jcm-10-01523-t004:** Adaptive behaviour of children and adolescents with KBG syndrome.

	Mean	Min–Max	SD	CI 95%
Full-Adaptive Scale	58.35	20–95	21.14	16.35–29.92
Communication domain	60.04	20–95	22.66	17.53–32.08
Daily Living Skills domain	66.17	22–98	20.96	16.21–29.66
Socialization domain	65.09	20–97	23.99	18.56–33.96

**Table 5 jcm-10-01523-t005:** Statistical differences between verbal comprehension index and the remaining indexes of Wechsler scales per each individual in our cohort compared to normative data of general population.

*N*	VCI vs. PRI		VCI vs. WMI		VCI vs. PSI	
	Delta	*p* Value	Delta	*p* Value	Delta	*p* Value
1	−9	0.26	5	0.37	13	0.24
2	31	0.01 **	33	0.02 *	29	0.06
3	12	0.20	3	0.42	−3	0.44
4	22	0.06	14	0.19	4	0.41
5	9	0.26	12	0.22	23	0.10
6	−12	0.20	−1	0.48	9	0.31
7	7	0.32	31	0.02 *	−5	0.39
8	−16	0.13	8	0.31	2	0.40
9	31	0.01 **	23	0.08	11	0.25
10	−6	0.34	−9	0.28	−12	0.25
11	34	0.01 **	16	0.15	27	0.07
12	12	0.20	0	0.50	−1	0.48
13	5	0.36	30	0.03 *	−6	0.37
14	18	0.10	13	0.20	10	0.29
15	−13	0.18	12	0.22	11	0.27
16	50	0.00 ***	36	0.01 **	68	0.01 **
17	−14	0.16	21	0.09	21	0.13
18	18	0.10	−12	0.22	−3	0.44
19	25	0.04 *	27	0.04 *	33	0.04 *
20	4	0.39	6	0.35	−6	0.37
21	−6	0.34	6	0.35	10	0.29
22	13	0.18	14	0.19	19	0.15
23	2	0.44	0	0.50	5	0.39
24	−15	0.14	−19	0.12	−13	0.24

* *p* < 0.05; ** *p* ≤ 0.01; *** *p* ≤ 0.001.

## Data Availability

The data that support the findings of this study are available on request from the corresponding author. The data are not publicly available due to privacy or ethical restrictions.

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
