# Peer review of "Cognitive and Adaptive Characterization of Children and Adolescents with KBG Syndrome: An Explorative Study"

_jcm, 2021, doi:10.3390/jcm10071523_

Round 1

Reviewer 1 Report

This is a nice study with a relevant cohort size of individuals with KBG-syndrome and as such important. The outline is clearly stated and assessment of functioning is done both on intellectual functioning and adaptive functioning. I do have few points of criticisms;

the prevalenve of KBG is much higher then the 1/1000000; authors should base their numbers on findings in lare sequencing studies

  • Methods – participants

There are no data reported on relevant medical features seen  in KBG and potentially could have influenced the results (epilepsy, medication, hearing and vision problems, hypotonia, etc). Authors should include those

  • statistic analyses –

It seems remarkable that a significant difference is measured between the indexes, because looking at the group means and SDs (table 3), differences between VCI and the other indexes are very small (even smaller than the SD of indexes themselves). I therefore doubt if a one-way ANOVA is the most suitable statistic tool to address the research question. In the analyses it is seen in df many more participants are counted then actually are recruited in the cohort (4x24=96). As it is a within-subject design, a choice for a repeated measures analyses would be better fit. As an alternative, a probably clinical more relevant approach would be to count the statistical differences between the indexes (Wechsler schales) and subschales (Vineland) per individual on the basis of signficance tables from the testguidances. In the results section an overview of frequencies can be given that show which indexes by how many individuals are different. In the conclusion it then can then be described how many individuals do actually have a relatively strong VCI and what the clinical implications might be.

Reviewer 2 Report

KBG is a distinct condition caused by mutations in ANKDR11. In recent years the widespread use of NGS techniques has helped to identify an increasing number of patients and shown, at the same time, the variable degree of cognitive abilities between them.

The identification of strengths and weaknesses in the cognitive, behavioural and social skills in individuals with specific syndromic conditions is crucial to design guided interventions in order to improve their intellectual and overall outcome.

Very little is known about KBG, so the authors set out to assess and delineate the cognitive and adaptive functioning profiles of 24 children and adolescents with a molecularly confirmed diagnosis of KBG syndrome by means of the corresponding widely used and validated scales.

The results showed a wide variability of intellectual levels but with strong verbal comprehension abilities compared to other domains (perceptual reasoning, working memory and speed processing). No differences were identified between the different domains of adaptive behaviour (communication, daily living skills and socialization).

This study has identified a specific pattern, with strengths in cognitive profile and weaknesses in adaptive functioning, in patients with KBG.

It is a relevant paper, with a sound methodology, very well structured and written, and with meaningful results.

Comments It would be useful to indicate in table 1 the two patients with a microdeletion (i.e. patients 10 and 18), and maybe also to clarify in the 3.1. Molecular Results section that these were very small deletions (288 and 275Kb, respectively) that include almost only ANKRD11 (size 205Kb). This is particularly relevant as next to ANKRD11, proximally, lies ZNF778, a candidate gene for autism, not included in neither of the two microdeletions. Moreover, both patients with microdeletions scored higher in both assessments than many patients with a point deletion.

I have no other comments to make.

Author Response

This manuscript is a resubmission of an earlier submission. The following is a list of the peer review reports and author responses from that submission.